# ViCor: Bridging Visual Understanding and Commonsense Reasoning with Large Language Models

## Abstract

In our work, we explore the synergistic capabilities of pre-trained vision-and-language models (VLMs) and large language models (LLMs) for visual commonsense reasoning (VCR). We categorize the problem of VCR into visual commonsense understanding (VCU) and visual commonsense inference (VCI). For VCU, which involves perceiving the literal visual content, pre-trained VLMs exhibit strong cross-dataset generalization. On the other hand, in VCI, where the goal is to infer conclusions beyond image content, VLMs face difficulties. We find that a baseline where VLMs provide perception results (image captions) to LLMs leads to improved performance on VCI. However, we identify a challenge with VLMs' *passive* perception, which often misses crucial context information, leading to incorrect or uncertain reasoning by LLMs. To mitigate this issue, we suggest a collaborative approach where LLMs, when uncertain about their reasoning, *actively* direct VLMs to concentrate on and gather relevant visual elements to support potential commonsense inferences. In our method, named **ViCor**, pre-trained LLMs serve as problem classifiers to analyze the problem category, VLM commanders to leverage VLMs differently based on the problem classification, and visual commonsense reasoners to answer the question. VLMs will perform visual recognition and understanding. We evaluate our framework on two VCR benchmark datasets and outperform all other methods that do not require in-domain supervised fine-tuning.

## 1 Introduction

The problem of visual commonsense reasoning (VCR) (Zellers et al., 2019; Hessel et al., 2022; Schwenk et al., 2022) expands upon the traditional visual question answering (Antol et al., 2015; Goyal et al., 2017). VCR requires machines to utilize commonsense knowledge for drawing novel conclusions or providing explanations go beyond the explicit information present in the image. To solve these problems, existing state-of-the-art methods mainly treat VCR as an image-text alignment task between the image content and candidate commonsense inferences (Hessel et al., 2022; Zhang & Fernando, 2023). These approaches, however, lack explicit modeling of the underlying reasoning steps, limiting their ability to generalize beyond the training data distribution. Recent methods have also been leveraging large language models (LLMs) for VCR problems (Hu et al., 2022; Shao et al., 2023).

However, these methods have several drawbacks. Firstly, they all require supervised training or fine-tuning on each specific dataset. Since different visual commonsense reasoning datasets have different focuses and data distributions (*e.g.* human-centric reasoning (Zellers et al., 2019) and reasoning in general topics (Schwenk et al., 2022)), the trained models struggle to generalize effectively to different datasets. Secondly, current state-of-the-art methods on different datasets are either based on supervised VLMs (Zellers et al., 2019; 2021) or combine VLMs (fine-tuned on in-domain VCR datasets) with LLMs (Hu et al., 2022; Shao et al., 2023). Notably, to the best of our knowledge, there is no comprehensive discussion on how VLMs and LLMs compare in the context of VCR problems and how to best harness their complementary capabilities.

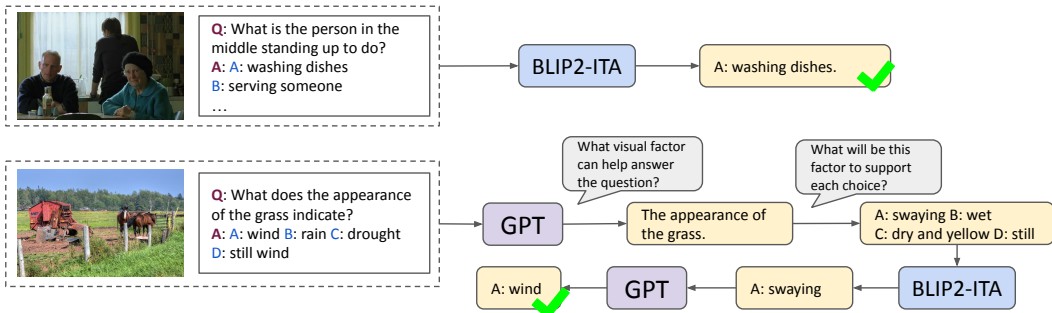

Figure 1: Two examples demonstrating different kinds of visual commonsense reasonings require different model capabilities. **Upper: Visual commonsense understanding (VCU)** requires the model to understand high-level concepts and attributes such as actions, events, relations, *etc*, which pre-trained VLMs can achieve via image-text alignment (ITA). **Lower: Visual commonsense inference (VCI)** requires the model to generate conclusions or explanations based on input image. Overlooking visual clues can result in erroneous conclusions. LLMs steer VLMs in discovering vital visual cues for answer support. The LLM employs the top ITA-scored visual clue (e.g.,"It is cloudy.") to perform commonsense inference.

To address these issues, in this work, as shown in Figure 1, we first systematically study the problem of visual commonsense reasoning and categorize it into two sub-problems – visual commonsense understanding (VCU) and visual commonsense inference (VCI). The *visual commonsense understanding (VCU)* problem requires the model to recognize various low-level visual patterns and then understand high-level concepts like actions, events, and relations in the image. The *visual commonsense inference (VCI)* problem requires the model to deduce conclusions or form explanations, likely to be true, based on visual observation. It requires a broad array of commonsense knowledge about the world, including cause-and-effect relationships, intentions, and mental states (Sap et al., 2020).

As prior work does not adopt this categorization, we instruct LLMs to classify these tasks, providing problem type descriptions along with a limited number of manually-annotated in-context samples. Based on the categorization, we assess the performance of VLMs (Li et al., 2023) and LLMs (equipped with image captions from VLMs) on VCU and VCI. Our findings (in Table 1) show that VLMs perform slightly better in VCU tasks, while also being more efficient. Conversely, in VCI tasks, LLMs outperform VLMs in most cases. This observation aligns with previous findings indicating that LLMs excel in text-based commonsense benchmarks (Anil et al., 2023).

We observe that image captions provided by VLMs such as (Li et al., 2023), often lack crucial contextual information necessary for answering questions. This poses a particular challenge for commonsense inference problems, as inferences are often defeasible given additional context (Choi, 2022). To illustrate this issue, consider the example depicted in Figure 1 (bottom). At first glance, it may appear that there's nothing noteworthy beyond horses on a grassy farm, leading one to select "D: still wind" as an answer. However, upon closer examination of the swaying grass, we must revise our conclusion to "A: wind." Existing perception modules, including VLMs, operate in a feed-forward manner and cannot adjust their perception based on a high-level understanding or inference. To address this, we propose instructing LLMs to intervene with VLMs in cases where they are uncertain about inference, typically indicative of a lack of sufficient visual evidence. This intervention would guide VLMs to focus on specific *visual factors*, such as weather or emotions, to support commonsense inferences.

We propose the **ViCor** framework, which employs the following components: (1) LLMs functioning as problem type classifiers (VCU and VCI), VLM commanders for directing VLMs based on problem classification, and visual commonsense reasoners to harness their extensive world knowledge and reasoning capabilities. (2) Pre-trained VLMs are responsible for visual recognition and understanding. Communication between LLMs and VLMs occurs through text, such as image captions, as they are universal medium for all existing models. On VCR (Zellers et al., 2019) and A-OKVQA (Schwenk et al., 2022), our method achieves state-of-the-art results among methods *without* supervised in-domain fine-tuning. On A-OKVQA, the result of ViCor is close to supervised state-of-the-art methods.

## 2 RELATED WORK

**Visual Commonsense Reasoning** Visual Commonsense Reasoning (VCR) Zellers et al. (2019); Hessel et al. (2022); Schwenk et al. (2022) is an emerging research area that aims to endow AI models with a human-like understanding and reasoning of visual scenes, beyond what can be directly observed. The goal is to understand high-level concepts such as events, relations, and actions and infer unobservable aspects such as intents, causal relationships, and future actions, requiring the integration of visual perception, understanding, and commonsense knowledge. The VCR task was introduced by Zellers et al. (2019), where models must answer a question about an image, given a set of four possible answers. Further, more datasets focus on more types of reasoning were proposed Park et al. (2020); Hessel et al. (2022); Schwenk et al. (2022). Most state-of-the-art methods treat VCR as an image-text alignment problem, where they encode the commonsesense inference and the visual input, then predict the alignment score of the image-text pair via a classification head or image-text similarity Zellers et al. (2019); Chen et al. (2020); Zellers et al. (2022); Hessel et al. (2022). Although achieving impressive performance, the generalizability of these methods is limited by supervised training. Recently, several works have leveraged large language models for visual commonsense reasoning Hu et al. (2022); Shao et al. (2023); You et al. (2023). However, Hu et al. (2022); Shao et al. (2023) require some VLMs trained on the datasets to provide visual information. You et al. (2023) use LLMs to decompose the main problem and use VQA models to acquire visual information. Our work systematically studies the visual commonsense reasoning problem and better leverages the strength of different pre-trained VLMs and the reasoning abilities of LLMs, which can be generalized to different visual commonsense reasoning datasets.

**Large Language Models for Vision-and-Language Tasks** Benefiting from the rich knowledge in LLMs, they have been used for various vision-and-language tasks in a zero-shot or few-shot manner. Yang et al. (2022); Hu et al. (2022); Shao et al. (2023) leverage LLMs for OK-VQA task Marino et al. (2019) by feeding the caption, question, candidate answers by VQA models, etc. to GPT3 models, and prompt the GPT model to answer the question with its pre-trained knowledge. Wang et al. (2022b) propose to use LLMs with image descriptors for video-language tasks. More recently, with the discovery of LLMs' tool using ability Yao et al. (2023); Schick et al. (2023), LLMs were equipped with various visual tools Gupta & Kembhavi (2023); Dídac et al. (2023); Shen et al. (2023); Lu et al. (2023); Wu et al. (2023) and achieved significant performance in Compositional Visual Question Answering, Science Question Answering tasks Suhr et al. (2018); Hudson & Manning (2019); Lu et al. (2022). Different from these works, we study a more complex and challenging task with different levels of reasoning, including requiring reasoning beyond direct image observation. In our method, the LLMs will perform reasoning for problem classification, visual information query, and commonsense reasoning.

## 3 VISUAL COMMONSENSE REASONING

### 3.1 VISUAL COMMONSENSE UNDERSTANDING

The visual commonsense understanding (VCU) problem requires the model to judge if a text $T$ describing a concept or an attribute aligns with the image $I$:

$$e = F(I, T) \tag{1}$$

where $e$ stands for evaluation of $T$ by model $F$. To answer these questions, the model needs to be able to map the low-level visual observations, such as objects and spatial relations to various high-level visual concepts and attributes, such as landmarks, actions, events, and relations.

### 3.2 VISUAL COMMONSENSE INFERENCE

The visual commonsense inference (VCI) problem usually requires the model to evaluate the plausibility of an inference about the image. Besides understanding the literal content in the image as in VCU, evaluating the inferences $T$ in VCI problems needs involves drawing novel conclusions or explanations from these visuals, often using (non-visual) commonsense knowledge and rules, based on some visual observations $\{o_i\}$ derived from the image:

$$e = F(\{o_i\}, T) \tag{2}$$

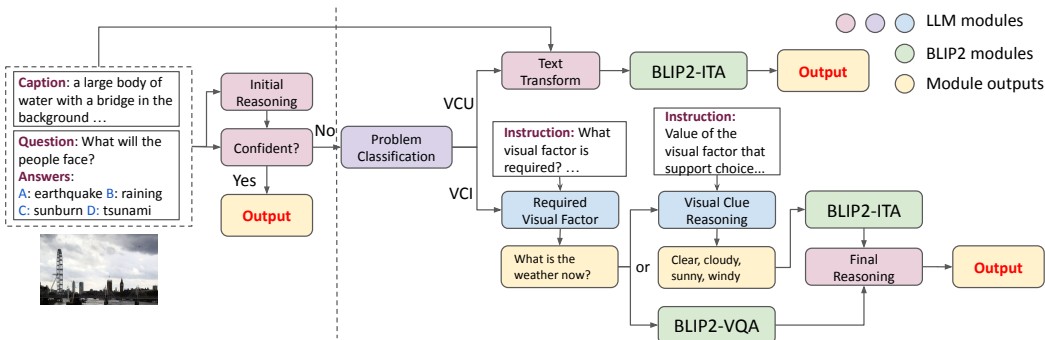

Figure 2: **Our ViCor framework.** Given a visual commonsense reasoning problem and a caption, our framework will leverage LLM to perform initial reasoning and confidence check. If the reasoning is not confident, the LLM will perform problem classification and acquire visual information according to the problem type. ∗Note that the final reasoning takes the question and the caption as input as well.

Here, $o_i$ could be some low-level visual observations or high-level visual commonsense understanding. Examples of non-visual commonsense knowledge could be the purpose of an object, people's opinions about an object, potential future events, etc.

### 3.3 VISUAL COMMONSENSE REASONING FORMULATION

Both categories of visual commonsense reasoning tasks share a common formulation. In visual commonsense reasoning, the input consists of two parts: an image denoted as $I$ and a multiple-choice question input represented as $q, c_i$, where $q$ corresponds to the question, and $c_i$ stands for the $i$-th answer choice. The model needs to choose the choice $c_i$ that is most likely to be true based on the image $I$.

## 4 METHOD

As shown in Figure 2, our approach involves a multi-step process. Initially, a pre-trained large language model (LLM) first takes the initial perception result (*i.e.*, image caption), a question-answer pair, and instructions as input to evaluate potential answer candidates. Then, if the LLM is not confident about its reasoning, it will reason about what visual factors should be perceived from the image to make a confident commonsense inference. Using this information as a guidance, a vision-and-language model (VLM) will focus on specific aspects of the image, returning the perception result back to the LLM. Finally, the LLM re-evaluates the candidates in light of new perception results.

### 4.1 LARGE LANGUAGE MODELS AS VCR REASONER

Evaluating answer choices VCR requires drawing new conclusions based on commonsense knowledge, which LLMs excels at (Anil et al., 2023). On the other hand, pre-trained vision-and-language models have exhibited a capability for visual understanding, including tasks such as image captioning and image-text alignment, with a demonstrated ability to generalize across various datasets, as highlighted in (Li et al., 2023). Therefore, we decided to harness the strengths of vision-and-language models for visual understanding and the capabilities of large language models for evaluating answer candidates in the context of visual commonsense reasoning.

Captioning serves as a fundamental unsupervised pre-training task and most generalized capabilities of pre-trained VLMs, which captures the most salient information from an image. Moreover, considering that text serves as a universal interface for both VLMs and LLMs, employing image captions serves as an effective means to connect VLMs with LLMs without necessitating any model-specific fine-tuning. Therefore, we first prompt the LLMs to take the caption of the image $C_I$ as the initial information and perform chain-of-thought reasoning on the question:

$$r_1 = LLM(\{c_i\}, q, C_I). \tag{3}$$

```
Please classify the question into one      You will be provided with a visual         You are given
of the following categories.               commonsense reasoning question about an     1. A main question and four choices for
1. Visual commonsense understanding:       image and four candidate choices.           an image.
questions that require understandings      Your task is to analyze what visual         2. A question about a visual factor for
of the image's current visual status.      factors are needed to evaluate the          the image.
2. Visual commonsense reasoning:           choices. Then, list the factors you         Your task is to think about an answer
questions require some visual              need in the format of a list of             for the visual factor question to
understanding of the image, then need     questions.                                  support each choice in the main
some commonsense knowledge to reason                                                   question.
about the answer.                          Examples:
                                           Question: Where is the plant on the
Examples:                                  sign usually found?                         Examples:
Question: What animal will most likely     Choices: A. desert B: tropics ......        Question: Where is the plant on the
eat this meal?                                                                         sign usually found?
Choices: A: elephant B: human ......       Analysis: To answer the question, we        Choices: A. desert B: tropics ......
                                           need to know the category of the plant      Visual factor question: What is the
Analysis: This question requires           on the sign.                                category of the plant on the sign?
understanding what is the meal in the      Required visual factors:
image, then using commonsense to reason    What is the category of the plant on        Reasoned answers:
which animal likes to eat it.              the sign?                                   A: cactus B. rainforest ......
......                                      ......
```

Figure 3: Three simplified prompt examples demonstrating how we define prompts to classify the problem (**left**), reason visual factors (**middle**), and think about visual observations regarding visual factors (**right**).

The reasoning result $r_1$ includes both intermediate reasoning steps and the final answer. However, it's important to note that the image caption may not encompass all the relevant information within the image, potentially omitting critical contextual details essential for answering the question. In such cases, it becomes necessary to gather additional relevant visual observations from the image. Before this, we must first judge whether there is a lack of supportive visual evidence that would allow us to make a confident decision. As in Figure 2, we let the LLM take the initial reasoning $r_1$ and the history prompt as input to judge if current visual information adequately supports the decision. If it does, the model will directly output the result. Conversely, if there is a lack of sufficient evidence, the model will progress to the second stage, where it will seek additional visual evidence.

## 4.2 Large Language Models as VCR Problem Classifier

As defined in the last section, there are two kinds of VCR problems, each requiring different levels of visual reasoning. Therefore, we propose to leverage VLMs in distinct manners when facing different problem types. To this end, we first prompt the LLM to classify the problem into two categories. To achieve this, we provide the definitions of these two categories in the prompt. Additionally, we include a set of manually annotated in-context examples to aid in problem classification, where the questions of in-context examples are selected from the training set. Figure 3 illustrates the prompt.

## 4.3 Large Language Models as VLM Commander

The pre-training dataset of vision-and-language models contains millions to billions of image-text pairs. Therefore, we propose a hypothesis that vision-and-language models have learned the mapping between visual features and the high-level commonsense concept during the pre-training. In light of this, for *visual commonsense understanding (VCU)* problems, we propose to leverage pre-trained VLM in a zero-shot manner. Specifically, for each choice $c_i$, we first instruct the LLM to transfer it and the question to a declarative sentence with instruction and in-context examples:

$$s_i = LLM(q, c_i) \tag{4}$$

For instance, for the question `What will the people face?` and the choice `earthquake`, we will transform them to `The people will face earthquake`. Then, we feed $s_i$ and the image $I$ to the pre-trained VLM to calculate the image-text alignment score. Following (Li et al., 2023), we use the sum of ITM and ITC scores to compare choices:

$$S_i = ITM(I, s_i) + ITC(I, s_i) \tag{5}$$

We will directly take the choice with the highest score as the final output.

For the *visual commonsense inference (VCI)* problems, the model needs to acquire related visual observations and use relevant commonsense knowledge to reason about the answer. This knowledge often neglected in the descriptions of the image. Therefore, as in Figures 2 and 3, we first prompt the LLMs to think about some *visual factors* $f_j$ that influence the answer to the question, like 'the action of the person', 'the interaction between people', etc. Then, we could acquire the visual observation

of the visual factor in the image with a visual question-answering model by asking a question about the visual factor:

$$o_j = VQA(I, f_j) \tag{6}$$

where $o_j$ is the answer to the question which we call *visual clue*. However, although pre-trained VLMs have zero-shot visual question-answering (VQA) capabilities (Li et al., 2023), VQA datasets require human labeling and are not one of the pre-training tasks of VLMs. Therefore, the accuracy and the quality of zero-shot VQA may hinder the performance of the reasoning for the question. Furthermore, the answer of VQA does not consider the context of the main question and therefore may lack the most useful information. To this end, we further propose to prompt the LLM to reason about the potential instantiations of the visual factors that can support the choices as in Figure 3:

$$o_{ij} = LLM(f_j, c_i, q) \tag{7}$$

As an illustration, when $f_j$ is "category of the plant," the potential values for $o_{ij}$ may include specific plant names like "cactus." Then, we could leverage the image-text matching (ITM) and image-text contrastive (ITC) functions of pre-trained VLMs to select the observation that most align with the image among the observations for each choice $i$:

$$o_j = o_{jk} \text{ where } k = \arg\max_i \{ITM(o_{ij}, I) + ITC(o_{ij}, I)\} \tag{8}$$

Finally, we append the *visual clues* $\{o_j\}$ after the caption as extra information for LLM to perform final reasoning:

$$r_2 = LLM(\{c_i\}, q, C_I, \{o_j\}) \tag{9}$$

## 5 EXPERIMENTS

### 5.1 DATASETS

We evaluate our approach using two datasets focused on visual commonsense reasoning: VCR (Zellers et al., 2019) and AOKVQA (Schwenk et al., 2022). Both datasets formulate visual commonsense reasoning as 4-choice QA problems about an image, containing various visual commonsense understanding and inference problems. VCR dataset focuses on human-centric visual commonsense reasoning problems. In contrast, A-OKVQA dataset requires various commonsense knowledge about common objects and events in daily life. For A-OKVQA, we use the validation set with 1145 examples. For VCR dataset, we randomly sample 3000 / 26534 examples from the validation set for the ablation study, and sample 500 examples to compare with other methods due to the cost of GPT4. We divide the image from left to right into three bins and name the person depending on which bin they are located in when feeding text to VLMs and LLMs, similar to (You et al., 2023). The performance of both datasets is evaluated by accuracy.

### 5.2 IMPLEMENTATION DETAILS

In our experiments, we use GPT-3.5-turbo-0613 and GPT-4-0613 as the LLMs for reasoning. To ensure reproducibility, we set the temperature of the LLMs to 0. For image captioning, we employ LLAVA-7B-v1.1. Furthermore, we use the pre-trained BLIP2 model for image-text alignment and BLIP2-FlanT5 XL for visual question answering on both datasets. The number of in-context examples used in the prompts shown in Figure 3 is 6, 1, and 3, respectively. All the questions in the in-context examples are from training set.

### 5.3 BASELINES

To demonstrate the effectiveness of our proposed framework, we implement the following zero-shot baselines for comparison:

- **BLIP2-Pretrain** (Li et al., 2023): We use the pre-trained BLIP-2 model directly to perform image-text alignment on both datasets. On both datasets, we utilize GPT-3.5-turbo-0613 to transform the questions and choices into declarative sentences and feed them to the BLIP-2 model to calculate the image-text alignment score. We select the choice with the highest alignment score as the answer.

Table 1: Ablations on the effect of problem categorization and clue generation on VCR Zellers et al. (2019) and A-OKVQA Schwenk et al. (2022) datasets. We use GPT-3.5-turbo-0613 for LLM-based methods. *Orig means using the declarative sentences transformed by LLM (Eq.5). *Clue means using the clues generated by LLM for image-text alignment (Eq.10). All numbers indicate accuracy (%). "Conf" indicates the samples where the LLM-Caption baseline shows confidence in its initial reasoning, while "!Conf" indicates cases where it lacks confidence.

| Decision Model | Visual Info | AOKVQA | | | | VCR | | | |
| | | VCU | | VCI | | VCU | | VCI | |
| | | Conf | !Conf | Conf | !Conf | Conf | !Conf | Conf | !Conf |
| --- | --- | --- | --- | --- | --- | --- | --- | --- | --- |
| BLIP2-Pretrain | Orig* | 76.5 | 66.3 | 56.5 | 50.9 | 70.0 | 56.3 | 59.2 | 47.4 |
| | LLM Clue* | 74.4 | 63.0 | 60.2 | 56.1 | 70.6 | 56.7 | 63.3 | 49.2 |
| LLM | Caption | 78.9 | 55.1 | 85.2 | 50.9 | 75.3 | 46.6 | 65.3 | 41.9 |
| | Caption + VQA Clue | 77.5 | 56.2 | 82.4 | 54.9 | 75.9 | 51.9 | 65.3 | 47.3 |
| | Caption + LLM Clue | 79.2 | 65.6 | 81.5 | 64.2 | 72.9 | 58.1 | 57.1 | 52.9 |
| Num. of Examples | | 289 | 575 | 108 | 173 | 170 | 1779 | 49 | 1002 |

- **IdealGPT** (You et al., 2023): A concurrent method leveraging LLMs for visual reasoning. IdealGPT prompts LLMs to iteratively query a VQA model to answer questions for visual reasoning tasks, including VCR (Zellers et al., 2019). In our experiments, we employ the original source code of IdealGPT while utilizing the *same* version of LLM and VLMs for caption, VQA, and reasoning as our method.

# 6 RESULTS AND ANALYSIS

## 6.1 ABLATION STUDY

We conduct ablation studies of our ViCor method on VCR and AOKVQA datasets. Results are shown in Table 1.

**How do VLM and LLM compare on visual commonsense reasoning?** By comparing the first row and the third row in Table 1, we can validate our hypothesis on the comparison between VLM and LLM. We observe that, in VCU problems, the VLMs perform significantly better than LLM reasoning based on the caption on both datasets, with an average accuracy of 63.6% vs. 56.0%. While on VCI problems, LLM based on caption performs better on average at 53.6% vs. 50.5%. We could also observe that BLIP2 has a significant performance gap between the two kinds of problems while LLM performs similarly. The significant difference between the two models and two datasets also validates the effectiveness of the problem classification performed by LLM.

**How do visual factors and LLM clue reasoning help visual commonsense reasoning?** We validate the effectiveness of visual factors reasoning and LLM clue reasoning on both BLIP2-Pretrain and LLM-based decision paradigms. Here, we describe how we adapt the clue generation method (as in Eq. 7) for BLIP2-Pretrain decision paradigm: we first prompt the LLM to generate the required visual factors $f_j$, then generate visual clues $o_{ij}$ of these factors that can support each choice $i$. When applying the clues to BLIP2-Pretrain, we take the average of the image-text alignment scores within the same choice as the image-text alignment score for the choice $i$:

$$S_i = \frac{1}{n} \sum_j (ITM(I, o_{ij}) + ITC(I, o_{ij})) \quad (10)$$

where $n$ is the number of required visual factors determined by LLM. The choice with the highest score will be selected.

From Table 1, we can first find that visual factors and visual clues are less helpful in **VCU** problems. On VCU problems, besides directly taking the concept being asked by the original question as the visual factor. The model will also consider low-level visual features as visual factors for the question. For example, for the question `What is the event in the image`, and the choice `dinner`, the visual factor could be `objects in the image`, and the reasoned visual clues could be `plates with food on the table`.

On BLIP2-Pretrain, using clues for image-text alignment is not better than using the transferred declarative sentences. This validates that BLIP2 can already align visual features with different

Table 2: Comparison between ViCor and other methods on VCR Q→A task. * Results on full validation set. † CoT indicates the same setting as 'Caption' baseline in Table. 1: given caption and perform chain-of-thought reasoning.

|  | Method | Acc.(%) |
|---|---|---|
| Sup. | R2C Zellers et al. (2019) | 67.3 |
| | *MERLOT Zellers et al. (2021) | 79.4 |
| ICL | BLIP2-Pretrain Li et al. (2023) | 51.2 |
| | *GPT-3.5* | |
| | †CoT | 43.8 |
| | IdealGPT You et al. (2023) | 47.9 |
| | ViCor (ours) | 55.4 |
| | *GPT-4* | |
| | CoT | 57.8 |
| | ViCor (ours) | 59.8 |

Table 3: Comparison between ViCor and other methods on A-OKVQA dataset. *Both PromptCap and Prophet trained VLMs on A-OKVQA dataset as part of the module. Sup. indicates supervised methods, and ICL means methods using in-context learning.

|  | Method | Acc.(%) |
|---|---|---|
| Sup. | GPV-2 Kamath et al. (2022) | 60.3 |
| | *PromptCap Hu et al. (2022) | 73.2 |
| | *Prophet Shao et al. (2023) | 76.4 |
| | InstructBLIP Dai et al. (2023) | 81.0 |
| ICL | BLIP2-Pretrain Li et al. (2023) | 65.6 |
| | *GPT-3.5* | |
| | CoT | 63.3 |
| | ViCor (ours) | 70.9 |
| | *GPT-4* | |
| | CoT | 70.3 |
| | AssistGPT Gao et al. (2023) | 74.7 |
| | ViCor (ours) | 75.6 |

concepts well. However, introducing visual factors and observations as extra context improves performance on LLM reasoning, especially when the LLM is not confident about its initial judgment, i.e., initial provided visual information (caption) is insufficient. In this case, the performance of LLM reasoning ('Cap + Clue' in Table 1) is comparable with pre-trained BLIP2.

For **VCI** problems, visual factors and visual clue generations help both reasoning paradigms. First, the improvement in the BLIP2-Pretrain paradigm validates that (1) pre-trained BLIP2 cannot well-align statements that go beyond literal visual content, requiring commonsense inference; (2) LLM can reason about the visual factors that may contribute to supporting candidate commonsense inferences, and guide the VLM to focus on relevant factors accordingly.

Second, the improvement in the LLM reasoning paradigm shows that LLM clues successfully provide subtle details of the scene that are crucial for solving the problem. Third, visual clues reasoned by LLM are better than VQA as the visual information provider. There are mainly two reasons. First, the pre-trained VLM sometimes could not understand or correctly answer the question due to the lack of language alignment. Second, the VQA model lacks the main question as the context and may not get the intention of the visual factor. Therefore, it may produce irrelevant answers. We provide examples to further illustrate these in Section 6.3.

**How to determine the reasoning process based on confidence and problem category?** When deciding the reasoning process, we need to consider both the performance and efficiency, evaluating by the number of LLM calls. From Table. 1, we can observe that when the LLM is confident about its initial reasoning, the performance is the best or almost the best on both VCU and VCI problems. Therefore, using LLM+caption is the best choice. When the LLM is not confident about its initial reasoning on VCI problems, LLM+Caption+LLM clue significantly outperforms other decision paradigms. On VCU problems, we can observe that the performance of BLIP2 is similar to LLM+Caption+LLM clue. However, the LLM+Caption+LLM clue requires five LLM calls, which is three times more than using BLIP2. Therefore, using BLIP2-ITA is the best choice in this case.

## 6.2 MAIN RESULTS

**VCR** The results on VCR dataset are in Table 2. Our method achieves the best result compared with other methods without supervised training. Specifically, our method outperforms the Ideal-GPT (You et al., 2023) since it is able to leverage the visual recognition and understanding abilities of VLMs more effectively by considering the types and definitions of problems. However, we notice that there is still a significant gap between ICL methods and methods with supervised training. This could be due to the loss of information in approximating the naming and labeling of the persons mentioned in Section 5.1.

**A-OKVQA** On A-OKVQA dataset, on both GPT models, our method can improve on chain-of-thought baseline by a significant margin. Compared with concurrent method AssistGPT (Gao et al., 2023), which utilizes GPT4 to call more visual tools such as object detection (Liu et al., 2023), text

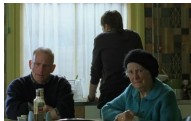 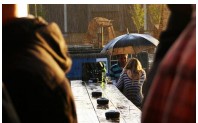 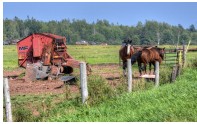

| Question | What is person in the middle standing up to do? | Why is woman holding umbrella? | What is the appearance of the grass indicating? |
|---|---|---|---|
| Choices | A: washing dishes B: serve someone | A: block sun B: repelling rain C: to dance | A: wind  B: rain C: drought D: still wind |
| LLAVA Caption | A man and a woman sitting at a table in a restaurant. The man is holding a bottle. | A group of people sitting at a table outside, with a woman sitting at the end under an umbrella. | A field with a tractor, a horse, and a cow. The tractor is parked in the grass. |
| Initial Evaluation | Given it's in a restaurant and the man is holding a bottle, it is likely that he is serving someone. | Since the caption does not mention any bad weather such as rain, the umbrella is to block sun. | The animals may not stand in rain or drought. The grass seems not moving – no strong wind. |
| Category | VCU | VCI | VCI |
| BLIP2-ITA results | A | A | D |
| Visual factors | N/A | The weather | Appearance of the grass |
| LLM clues | N/A | A: sunny B: rainy C: sunny | A: sway B: wet C: wither D: motionless |
| VQA result | N/A | rainy | It is a grassy field |
| Final result | A: washing dishes | B: repelling rain | A: wind |
| Ground truth | A | B | A |

Figure 4: **Qualitative examples.** All the examples are in the case of initial reasonings are not confident. **Left:** An example in the **VCR** dataset, where the ITA corrects the initial reasoning. **Middle:** An example in the **A-OKVQA** dataset, where the LLM corrects the initial reasoning after giving the observation of the visual factor. **Right:** An example in the **A-OKVQA** dataset, where the reasoned clue provides more useful information than VQA.

detection, and region grounding (Wang et al., 2022a), our method with only BLIP2 and LLAVA can achieve better results. Meanwhile, we can observe that our method ViCor, without any training on the dataset, can achieve results close to the best supervised methods. This shows that our analysis and modeling for visual commonsense reasoning makes our framework tackle the VCR problems more efficiently.

## 6.3 QUALITATIVE EXAMPLES

In Fig. 4, we demonstrate several qualitative examples. The left example shows a case where the problem is classified as VCU, and the BLIP2-Pretrain selects the correct answer. The middle example presents a case where the initial evaluation is incorrect, and both the VQA and clue reasoning methods give the correct observation for the visual factor 'weather', based on which the LLM selects the correct answer. The BLIP2-Pretrain here selects 'block sun' due to the lighting condition of the image. The example on the right demonstrates a case when the LLM reasoned answer is better than the answer generated by the VQA model. Here, the VQA does not understand the intention of the visual factor without the context of the main question. The LLM reasoned answer, however, can provide the most relevant information to the question and help the final reasoning. The BLIP2-Pretrain fails here due to the textual similarity between 'wind' and 'still wind'.

## 7 CONCLUSION AND LIMITATIONS

In this work, we study the problem of visual commonsense reasoning (VCR) based on the capabilities of pre-trained vision-language models and large-language models and define two sub-problems – visual commonsense understanding (VCU) and visual commonsense inference (VCI). Based on this, we propose the ViCor framework that efficiently uses visual recognition and understanding capabilities of VLMs and commonsense reasoning capabilities of LLMs to overcome the challenges in VCR. The experiment results validate our analysis of VCR problems and the effectiveness of our framework. Currently, our method lags behind best performing methods in the field which are based supervised fine-tuning. Also, text is the only communication medium between LLMs and VLMs. The loss of visual details caused by captions may be hindering on certain scenarios. Future work could explore fine-tuning approach with alternative mediums such as visual embeddings.

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

Table 4: Ablations on VCR with more decoding configurations.

| Decision Model | Decoding Config | VCU | | VCI | |
|---|---|---|---|---|---|
| | | Conf | !Conf | Conf | !Conf |
| LLM + Caption + LLM Clue | Orig | 72.9 | 58.1 | 57.1 | 52.9 |
| | Temp 0.1 | 72.4 | 58.8 | 57.1 | 53.9 |
| | Temp 0.2 | 72.4 | 58.5 | 61.2 | 52.2 |
| | ICL examples | 74.7 | 56.7 | 59.2 | 54.2 |
| Num. of Examples | | 170 | 1779 | 49 | 1002 |

Table 5: The result of ViCor on OKVQA dataset.

| Method | Accuracy |
|---|---|
| LLM+Caption | 34.3 |
| BLIP2-T5XL | 36.0 |
| ViCor (ours) | **38.4** |

# A  ADDITIONAL RESULTS

## A.1  RESULTS ON MORE LLM DECODING CONFIGURATIONS

To validate the robustness of our method, we ran the experiments on the VCR dataset with more decoding configurations using LLM + Caption + LLM Clue decision branch. Specifically, we ran on two more LLM decoding temperatures 0.1 and 0.2, and used different in-context examples for the prompt in Fig.3 (right) to guide the LLM to think about observations for visual factors based on candidate choices. From the results in Table. 4, we can observe that different decoding configurations influence the results by a small margin and do not affect the main conclusions.

## A.2  RESULTS ON OKVQA DATASET

We adapt our method and baselines to OKVQA Marino et al. (2019) dataset. The results are in Table. 5. We use GPT-3.5-Turbo for LLM modules. Since OKVQA is an open-ended dataset, we use the Caption+VQA clue version of our method in Table 1 to tackle unconfident VCI problems. As shown above, our framework can still leverage the advantage of both VLMs and LLMs to achieve better results owing to problem classification and active visual information acquisition.

