# OpenReview forum: "ViCor: Bridging Visual Understanding and Commonsense Reasoning with Large Language Models"
_ICLR.cc/2024/Conference — Submitted to ICLR 2024_

### Official Review · Reviewer_ZNkb · 2023-10-17

**Soundness:** 1 poor
**Presentation:** 1 poor
**Contribution:** 1 poor
**Rating:** 3
**Confidence:** 4

**Summary:**

This paper dissects the visual common sense reasoning task into two sub-problems: visual common sense understanding and visual common sense inference.
A method that connects these two sub-tasks is proposed based on VLM and LLM.
The experiments are conducted on the VCR and A-OKVQA datasets.

**Strengths:**

- The studied problem - visual common sense reasoning, is useful and practical for evaluating large models' reasoning capabilities.
- The finding that the captions cannot be used for answering questions is interesting to know.

**Weaknesses:**

- The re-definition of visual common sense reasoning is not convincing at all.
There is a large overlap between the two sub-problems of visual common sense understanding and visual common sense inference.
Moreover, reasoning and understanding are also very close.

- Even with this new definition, the authors still perform their experiments on the VCR dataset, which is extremely confusing.

- There is no explicit definition of VCI.

- In fact, the definition of VCU is no different from that of image text retrieval.

- Fig.4, I think, should be changed with a table rather than drawing a table.

**Questions:**

See the weakness part.
Overall, I think this paper is far from the acceptance bar of ICLR.

---

> ### Author Response · Authors · 2023-11-21
>
> We would like to express our appreciation for recognizing the usefulness and practicality of our study. We are happy to address the concerns raised by the reviewer as below.
> 1. **Problem classification for Visual Commonsense Reasoning.** We would like to clarify that we didn’t redefine the VCR problem. VCU focuses on **perceiving and understanding literal** visual contents in an image, while VCI involves drawing novel conclusions or explanations from these visuals, often using (non-visual) commonsense knowledge and rules. The objective of this categorization is to leverage the strengths of LLMs in commonsense reasoning and VLMs in visual perception (i.e. understanding literal visual contents). The findings in Table 1, which reveal a significant contrast in the behaviors of VLMs and LLMs when it comes to VCU and VCI, support dividing VCR into VCU and VCI is an effective categorization.
>
> 2. **Why perform experiments on VCR datasets?** As mentioned in the 1st point, the problem we are studying is visual commonsense reasoning, and we validate the efficacy of our problem classification in Section 6.1.
>
> 3. **Definition of VCI.** We would like to kindly point out that the definition of VCI is in the Sec. 3.2 of the paper. We’ve made the distinction between VCI and VCU clearer in our revision.
>
> 4. **Relation between VCU and image-text retrieval.** We appreciate the reviewer's insight regarding the connection between VCU problems and image-text retrieval. While acknowledging this connection, we argue that this relationship is not a limitation of our work. As per the ICLR 2024 Reviewer instructions, “originality may arise from […] **creative combinations of existing ideas, application to a new domain**.” In this vein, our contribution is unique in that we are the first one to solve a subset of VCR, namely VCU, using image-text alignment capabilities in VLMs. We believe that our approach of categorizing problems is significant, as it has the potential to reduce costly API requests to LLMs.
>
> 5. **Reason for figure 4.** We appreciate the reviewer for the suggestion. However, we believe combining the example images and the analysis as a figure does not affect the presentation.

---

### Official Review · Reviewer_eVt8 · 2023-10-21

**Soundness:** 2 fair
**Presentation:** 3 good
**Contribution:** 2 fair
**Rating:** 5
**Confidence:** 5

**Summary:**

This paper proposes a vision-language joint reasoning framework, ViCor, to achieve the synergy between a pretrained vision-language model (VLM) and an LLM. ViCor has outperformed other prompting-LLM-based approaches on two visual commonsense reasoning benchmarks, namely VCR and A-OKVQA.

Concretely, ViCor involves using an LLM as a problem categorizer (high- vs. low-confidence; understanding vs. inference). Then the LLM also acts as a commander to query appropriate visual clues from the VLM. These manners of directing information flows according to the question property is demonstrated to be effective in harnessing complementary benefits of both VLM and LLM.

However, ViCor still underperforms supervised finetuning methods, which is left for future work.

**Strengths:**

This paper provides important insights about the comparative advantages of VLMs and LLMs, and introduces an effective framework where they can collaborate. Concretely, VLMs are better at recognizing literal visual content. Their contrastive pretraining has equipped them with strong image-text alignment capabilities. However, VLMs lack commonsense or world knowledge. Thus VLMs could be benefit from LLMs providing texts including meaningful visual clues to compute alignment scores. On the other hand, LLMs posses a wealth of  commonsense and world knowledge, and are better at expanding or decomposing problems. But LLMs do not have direct access to visual information, thus would require a VLM to act at their commands.
Studies in this paper have attempted to revealed that:
- VLM as the decision model is suboptimal due to the lack of overall reasoning ability, unless the task is as simple as recognizing the literal visual content.
- LLM as the decision model works only when it partners with a VLM and queries the VLM with **visual-clue**-rich texts rewritten from the original textual question.
- The collaborative paradigm between VLM and LLM mitigates mistakes originated from a) easily overlooked visual clues in the surroundings, b) lack of explicit mentions of relevant visual factors in the question, c) misdirecting objects in the foreground.

**Weaknesses:**

- Judging from Table1, the number of examples studied is very small. It is unclear if the evidence derived from the results is robust.
- No multiple runs across decoding configs (e.g. temperature, selection of in-context examples). This limits the robustness and generality of the findings.
- Judging from Table1's LLM section: I'm having a hard time drawing conclusive insights from these results. LLM+Caption wins in two columns, LLM+Caption+VQA wins in two columns, while LLM+Caption+LLMclue wins in 4 columns. None of the settings is consistently stronger. Nor do the results suggest a consistent way to choose settings based on the problem category.

I believe that demonstrating how LLM and VLM can wisely collaborate in reasoning tasks is a direction worth pursuing. Therefore, I don't question the motivation of this paper. However, this paper only produced preliminary results on small validation sets and with a single set of decoding configurations. So, the results might lack generality and comprehensiveness. More comprehensive experiments across larger datasets and multiple seeds/decoding configurations would significantly strengthen the arguments the authors have sought to put forth.

**Questions:**

- Sec5.2 "BLIP2-FlanT5 XL for visual question answering": I'm having a hard time understanding the LLM+VQA workflow. I didn't see a corresponding VQA module in Fig2. Moreover, the paragraph between Eq6&7 explained why a VQA model often faces diffifulties, so I thought you have deemed a VQA model inadequate. Could you clarify whether you include a VQA model due to its comparative advantages under certain circumstances, or simply as a baseline? Also, could you update Fig2 and show where a VQA model would fit in the workflow?
- Sec6.3 "VQA does not understand the intention of the visual factor without the context of the main question": Have you tried concatenating the original question and the question about the visual factor as joint inputs to the VQA model?
- Table2: What is ICL? (Image-contrastive learning?)
- It appears that the categorization of VCU/VCI is solely determined by the LLM. Have you considered any measure to empirically test the reliability of this classification?

---

> ### Author Response · Authors · 2023-11-21
>
> We sincerely appreciate the reviewer for their recognition of the insight of our study and the effectiveness of our proposed method. We are happy to address the concerns and questions raised in the review below:
> 1. **The rationale for branching based on the problem category and the confidence.** This is an important question. In the decision of choosing the branch for solving the question, we consider two aspects: performance and efficiency, evaluating by the number of LLM calls.
>
>     (a) From Table 1, we can observe that when the LLM is confident about its initial reasoning, its performance is superior or nearly the best for both VCU and VCI problems. In addition, [LLM w/ Caption] is the most compute-efficient. Therefore, using [LLM w/ Caption] is a **cost-effective** option.
>
>     (b) When the LLM is not confident about its initial reasoning on VCI problems, [LLM w/ Caption+LLM clue] significantly outperforms other decision paradigms. On VCU problems, we can observe that the performance of BLIP2 is similar to [LLM w/ Caption+LLM clue]. However, the [LLM w/ Caption+LLM clue] requires five LLM calls, which is three LLM calls more than using BLIP2. Therefore, using BLIP2-ITA is a **cost-effective** option in this case.
>
>     We briefly mentioned the efficiency aspect in the original version. We’ve updated the explanation for the branching rationale in Section 6.1.
>
> 2. **Results on more examples and decoding configurations of VCR dataset.** Thank you for the suggestion. We agree running on more examples and more decoding configurations could better help validate our findings. Therefore, we sample **3000 examples** from the VCR dataset. The results are shown below:
> | Model           | VCU conf | VCU !conf | VCI conf | VCI !conf |
> |-----------------|----------|-----------|----------|-----------|
> | BLIP2-Pretrain  | 70.0     | 56.3      | 59.2     | 47.4      |
> | BLIP2-Pretrain-clue | 70.6 | 56.7      | 63.3     | 49.2      |
> | Caption             | 75.3     | 46.6          | 65.3     | 41.9      |
> | Caption + VQA Clue | 75.9  | 51.9      | 65.3     | 47.3      |
> | Caption + LLM Clue | 72.9  | 58.1      | 57.1     | 52.9      |
>
>     As shown above, the conclusions drawn from Table 1, and the branching strategy explained above (and in the revision of the paper) remain unchanged compared with the results on 500 examples.
>
>     We also run the **[Caption + LLM Clue]** version in Table 1 with extra temperatures and with different in-context examples in the prompt for thinking visual observations to support each choice (Fig.3 Right).
>
>     | Caption + LLM Clue  | VCU conf | VCU !conf | VCI conf | VCI !conf |
>     |-----------------|----------|-----------|----------|-----------|
>     | Original        | 72.9     | 58.1      | 57.1     | 52.9      |
>     | temp 0.1        | 72.4     | 58.8      | 57.1     | 53.9      |
>     | Temp 0.2        | 72.4     | 58.5      | 61.2     | 52.2      |
>     | ICL example     | 74.7     | 56.7      | 59.2     | 54.2      |
>
>    We can also observe that, using different LLM temperature and in-context examples only have a small influence on the performance, and does not affect the conclusions. We’ve updated the results in the revision.
>
> 3. **Explanation of LLM+VQA workflow and update of Figure 2.** We are sorry for the confusion, as explained in Sec 4.3, in our framework, both the VQA module and LLM clue could be the source of extra visual information in the uncertain scenario. We only include the LLM clue pipeline in Fig. 2 due to its superior performance in our experiments. We’ve fit the VQA pipeline into Fig.2 in our revision.
>
> 4. **Concatenating the original question and the question about the visual factor as joint inputs to the VQA model.** Providing the VQA model in the context of the original question is a great idea that may solve the issue of the VQA model, and we thank the reviewer for bringing this out. We’ve implemented this method on VCR dataset and the result is as below:
> | ViCor (GPT 3.5)              | VCR  |
> |------------------------------|------|
> | Caption + VQA Clue           | 50.6 |
> | Caption + VQA Clue w/ context| 50.5 |
> | Caption + LLM Clue           | **57.3** |
>
>     The results are on the 3000 subset sampled in the response above. In the comparison, we can observe that this method doesn’t improve the performance compared with the original VQA prompt. This is because answering the visual factor question conditioned on the main question is a complex problem, which is never seen in the training of BLIP2 or the LLM used by BLIP2 – Flan T5 XL. Therefore, the BLIP2 still fails to provide more relevant visual information to the original question.
>
> 5. **Explanation of ICL.** We are sorry for the confusion. ICL means in-context learning, in which several human-defined examples are provided as demonstrations of the task to help the model perform a task without training. We’ve added an explanation of it in Table 3.

---

> ### Author Response · Authors · 2023-11-21
>
> 6. **Evaluation of the reliability of the problem classification by LLMs.**  The reliability of the problem classification can be reflected in how the LLMs and VLMs paradigms perform in the classified problems. First, as explained in the first paragraph in Section 6.1, the significant contrast between the performance of both LLM and VLM-based decision models on the classified subproblems demonstrates that the problems classified in different categories have significant differences. Second, the fact that treating the problems differently by problem category balances the performance and efficiency well also shows the effectiveness of the problem classification by LLM.
>
>     Since the definition of the two subproblems is not proposed in the original dataset creation, we could not evaluate the accuracy of the problem classification on a large scale. However, we randomly select 50 examples from AOKVQA dataset and manually label the category of them. The comparison between the LLM classification and human labeling is shown below:
>     |          | Human VCU | Human VCI |
>     |----------|-----------|-----------|
>     | **LLM VCU**  | 32        | 4         |
>     | **LLM VCI**  | 4         | 10        |
>
>     As we can see, the accuracy of the LLM classification is 84%, which shows that LLM’s classification has a high correlation with human’s classification.

---

### Official Review · Reviewer_iFin · 2023-10-29

**Soundness:** 3 good
**Presentation:** 3 good
**Contribution:** 2 fair
**Rating:** 6
**Confidence:** 4

**Summary:**

The paper suggests current visual-language model(VLM) often perceives the image passively and does not provide crucial contextual information to LLM for answering questions. Thus it category the commonsense VQA into two categories: visual commonsense understanding (VCU) and visual commonsense inference（VCI). For the harder VCI problems, the paper proposes to prompt LLM to generate questions and query VLM to obtain related visual information. The proposed method improves the baseline in-context learning performance on the knowledge-based VQA task.

**Strengths:**

The proposed method is well-motivated. Current pre-trained VLMs do not extract visual context based on the input questions. The proposed method can address this problem.

**Weaknesses:**

Existing methods, such as BLIP2, instructBLIP, and mini-GPT4, align the visual context with the LLM inputs embedding instead of input words and achieve better performance. It is not clear why the proposed method uses words(caption or VQA result) to transfer information from the VLM to LLM.

The paper lacks implementation details of the proposed model and compared methods. BLIP2 has a different setting to obtain the answer to the original one.

**Questions:**

- What is the j in o_j represent? How is the range of j determined?
- How are the in-context examples obtained? From training set or manually written?
- The BLIP-2 can directly generate the answer for visual questions. How does the BLIP-2 perform in this setting?
- The instructBLIP is not trained on the VCR dataset. What is it zero/few-shot performance on VCR?
- What is the LLM size of the compared baseline? Do they have a similar number of parameters to the proposed model?

---

> ### Author Response · Authors · 2023-11-21
>
> We sincerely appreciate the reviewer’s recognition of the motivation of the study and the proposed method. We are happy to address the concerns and questions raised in the review below:
> 1. **Why use words to transfer information between LLMs and VLMs.**
> This is an important question. We acknowledge that relying solely on text for communication between LLMs and VLMs is a limitation of our study, a point we also mentioned in the limitations section. However, we believe our submission is a valuable addition to the literature due to following reasons:
>
>     **(a)** Currently, connecting top-performing closed-source proprietary LLMs directly with perception (e.g. image, video, point cloud, etc.) modules through embeddings is challenging. As a result, the use of captions remains the only method for establishing this connection. Hence, most current research in multi-modal applications of LLMs primarily relies on caption-based approaches.
>
>     **(b)** Utilizing text for communication between LLMs and VLMs provides a clear advantage in terms of interpretability. This interpretability is crucial for comprehending the behavior of our method. While embeddings could serve as an alternative medium, they tend to act as a "black-box", offering limited interpretability. This lack of clarity can lead to challenges in debugging and elongate the development process.
>
>     **(c)** In our initial investigations, we observed a comparable effect when linking LLMs and VLMs through embeddings. Specifically, these models can neglect important visual cues in scenes characterized by clutter and occlusions because they do not actively search for visual cues based on the context of the scene and the question. One possible extension of our approach could involve implementing an attention mechanism over visual embeddings, guided by a high-level comprehension of the scene and contextual information, much like the methodology we have employed.
>
> 2. **Implementation details of experiments and BLIP2 with LLM.** Thank you for the suggestion. We included the implementation details of the compared methods and our method in Sec. 5.2 and Sec. 5.3. We also add the results of the implementation of BLIP2 with LLM below and in the paper.
> | Model        | AOKVQA | VCR  |
> |--------------|--------|------|
> | BLIP2-Pretrain | 65.6   | 51.2 |
> | BLIP2-T5xl    | 71.6   | 53.6 |
> | ViCor (ours)       | **75.6**   | **59.8** |
>
>     As shown above, the BLIP2 with LLM version performs better than the version without LLM, which is mainly because the LLM of BLIP2 improves the commonsense reasoning capability based on the perceived information. Equipping BLIP2 with larger LLMs could potentially improve its performance on visual commonsense reasoning. Currently, connecting top-performing proprietary LLMs like GPT-4 directly with BLIP2 through embeddings is not possible. As a result, the use of captions remains the only method for this connection, underlining the importance of our approach.
>
> 3. **Explanation of $j$ in $o_j$** As in Eq. 6, $o_j$ corresponds to the $j^{th}$ visual factor. The range of $j$, which is the number of visual factors, is determined by the visual factor reasoning performed by the LLM as in Fig. 2 (e.g. there could be one or multiple visual factors required to answer the question). The prompt for visual factor reasoning is in Fig. 3 middle.
> 4. **Construction of the in-context examples.** The questions in the in-context examples in the prompt are selected from the training set.
> 5. **Results of InstructBLIP on VCR.** Thank you for the suggestion. The structure of InstructBLIP doesn’t support few-shot **multi-modal** in-context learning. Therefore, we test the zero-shot result of InstructBLIP on VCR dataset, and the result is shown below.
> | Model             | VCR  |
> |-------------------|------|
> | InstructBLIP-Vicuna7b | 46.2 |
> | InstructBLIP-T5xl    | 54.6 |
> | ViCor (ours)            | **59.8** |
>
>     From the result, we can see that instruction tuning on a large number of standard vision-and-language benchmarks improves the generalization abilities of VLMs to new tasks compared with BLIP2 using the same T5-XL LLM. However, the performance of InstructBLIP-Vicuna7b version is significantly worse, and our ViCor method still significantly performs better.
>
> 6. **LLM size of the compared baselines.** The number of parameters in GPT-3.5-turbo or GPT-4 is unknown. However, concurrent in-context-learning-based methods and chain-of-thought baselines listed in Table 2 and Table 3 use the same LLMs as ours. BLIP2 and InstructBLIP compared above are using T5XL with 3B parameters, and Vicuna with 7B parameters.

---

> > ### Comment · Reviewer_iFin · 2023-12-02
> >
> > The additional experiments addressed most of my concerns. However, I am not fully convinced that the caption is necessary for bridging vision and language models. I will raise my rating to 6.

---

### Official Review · Reviewer_GM3o · 2023-10-29

**Soundness:** 2 fair
**Presentation:** 2 fair
**Contribution:** 2 fair
**Rating:** 6
**Confidence:** 4

**Summary:**

This paper explores using vision-language models and language models for visual commonsense reasoning.  The VCR problem is categorized into two parts: (1) visual commonsense understanding (VCU) and (2) visual commonsense inference (VCI).  The paper identifies that VLMs may struggle with VCI and therefore, the paper employs LLMs to aid and collaborate VLMs for better VCR.  Experimental results are demonstrated for two VCR datasets (VCR from Zellers et al. and AOKVQA from Schwenk et al.)

**Strengths:**

1. This paper provides strong empirical results when combining proprietary language models such as GPT with vision-language models for solving VCR problems.
2. I wouldn't want to dismiss the paper as "combination of multiple proprietary blackboxes" (although it probably is that) -- the paper does demonstrate that there are novel ways to leverage these tools for solving challenging problems in vision.
3. The paper is well written and well explained to someone who is already familiar with the advances in this domain. See Weakness 4 for the flip side.

**Weaknesses:**

1. Experiments could be more exhaustive -- for instance, why not expand the experiments into more VCR datasets such as OKVQA (Marino et al.), VisualComet (Park et al.), V2C (Fang et al)?
2. The pipeline doesn't seem to be specific to VCR and could be used for any VQA dataset (eg. VQAv2, GQA, CLEVR, etc.) -- it's not clear whether the proposed method also improve performance on these datasets.  In practice, questions to a real-time system could be of any type (those about commonsense or those about simple perception) -- so it would be important to improve performance on both.
3. In Table 1, it is unclear how each dataset is divided into two parts VCU and VCI for evaluation.
4. The paper is well written and well explained to someone who is already familiar with the advances in this domain, but this assumption could have limiting effects on who learns from the paper -- one of the advantages of publishing NLP papers in ICLR is a wider reach to the broad ML community (and a large part of this community does not work on NLP or LLMs).

**Questions:**

1. Please define "Sup" and "ICL" for readers -- the acronyms may be common in the active LLM research community, but you cannot assume that readers are part of this active community, especially since in-context learning is only a couple of years old.

---

> ### Author Response · Authors · 2023-11-21
>
> We would like to express our heartfelt gratitude to the reviewer for recognizing the novelty and strong results of our work. We would love to address the concerns raised in the review below.
>
> 1. **The design motivation of our method and experiments on more datasets.** Thanks to the reviewer for mentioning these VQA datasets. We agree that a system that can improve all kinds of visual question answering would be the best. However, the motivation of the design of our framework is twofold: firstly, to leverage complementary strengths of VLMs and LLMs under VCU and VCI problems; and secondly, to enable let LLMs direct VLMs in gathering relevant visual elements that aid in making commonsense inferences. Therefore, our study and framework does not aim to improve the **grounding and spatial perception capabilities**, which general visual question answering benchmarks like VQA and GQA emphasize.
>
>     As suggested, we apply our framework to another visual **commonsense reasoning** dataset, specifically OKVQA. It's important to highlight that OKVQA represents an **open-ended** question-answering (QA) benchmark, distinguishing it from the **multiple-choice** QA benchmarks presented in our original submission. We randomly sample 1000 examples from OKVQA dataset and test our framework on them, the results are shown below:
>     | Model     | OKVQA |
>     |-----------|-------|
>     | LLM+Caption | 34.3  |
>     | BLIP2-T5xl | 36.0  |
>     | ViCor (ours)    | **38.4**  |
>
>     Since OKVQA is an **open-ended** QA dataset, we use the Caption+VQA clue version of our method in Table 1 to tackle unconfident VCI problems. We use GPT-3.5 as the LLM modules in the experiments. As shown above, our framework can still leverage the complementary capabilities of both VLMs and LLMs to achieve better results owing to problem classification and active visual information acquisition.
>
> 2. **How to divide VCU and VCI problems in the experiments.** This is a good question. In the ablation study presented in Table 1, the division between VCU and VCI is based on the output from GPT-3.5-turbo. The prompt we used is shown in Figure 3. As shown in our response to Reviewer eVt8 (Evaluation of the reliability of the problem classification by LLMs), the problem category classification by LLMs aligns well with human categorization, ensuring reliability in identifying problem types. It is important to note that in real usages (results of Tables 2 and 3), LLMs will attempt to classify the question into either VCU or VCI problem **only if** it is not confident in the initial reasoning, as shown in Fig. 2.
> 3. **Clearer definitions.** Thanks to the reviewer for the warm suggestion. We’ve defined these terms in our revision.

---

### Author Response · Authors · 2023-11-21

We sincerely appreciate the reviewers for their constructive comments. Overall, the reviewers acknowledge that our method is novel (GM3o), well motivated (iFin,eVt8), effective (eVt8), and supported by strong empirical results (GM3o). The primary critiques relate to the limited scale of our experiments (GM3o, eVt8) and some minor ambiguities in our writing (GM3o, iFin, eVt8, ZNkb). We’ve made the following main changes to our paper according to the suggestions in the review and highlighted these changes.

1. We extend our framework to an **open-ended** visual commonsense reasoning dataset OKVQA, which is different from the **multiple-choice** benchmarks presented in our original submission. (GM3o Q1)
2. We perform **larger scale** experiments on VCR dataset, and validate the robustness of our framework with multiple decoding configurations. (eVt8 Q2)
3. Clearer explanation has been provided to

    (1) the core difference between the VCU and VCI problems. (ZNkb Q1)

    (2) how the ablation results in table 1 support the choice of reasoning process. (eVt8 Q1)

    (3) Update the figure 2 to fit both VQA clue and LLM clue pipelines. (eVt8 Q3)

    (4) The definitions of some terms.

---

> ### Author Response · Authors · 2023-11-23
>
> Below, we reiterate our answers to key questions raised by the reviewers.
>
> **[Results on an additional benchmark]**
>
> As suggested by Reviewer GM3o, we apply our framework to another visual **commonsense reasoning** dataset, specifically OKVQA. It's important to highlight that OKVQA represents an **open-ended** question-answering (QA) benchmark, distinguishing it from the **multiple-choice** QA benchmarks presented in our original submission. We randomly sample 1000 examples from OKVQA dataset and test our framework on them, the results are shown below:
>
> | Model     | OKVQA |
> |-----------|-------|
> | LLM+Caption | 34.3  |
> | BLIP2-T5xl | 36.0  |
> | ViCor (ours)    | **38.4**  |
>
>  Since OKVQA is an **open-ended** QA dataset, we use the Caption+VQA clue version of our method in Table 1 to tackle unconfident VCI problems. We use GPT-3.5 as the LLM modules in the experiments. As shown above, our framework can still leverage the complementary capabilities of both VLMs and LLMs to achieve better results owing to problem classification and active visual information acquisition.
>
>
> **[Results on a larger subset of VCR dataset]**
>
> As suggested by eVt8, we run our method on more examples and more decoding configurations to better validate our findings. Specifically, we sample **3000 examples** from the VCR dataset (500 examples were used in the initial submission). The results are shown below:
> | Model           | VCU conf | VCU !conf | VCI conf | VCI !conf |
> |-----------------|----------|-----------|----------|-----------|
> | BLIP2-Pretrain  | 70.0     | 56.3      | 59.2     | 47.4      |
> | BLIP2-Pretrain-clue | 70.6 | 56.7      | 63.3     | 49.2      |
> | Caption             | 75.3     | 46.6          | 65.3     | 41.9      |
> | Caption + VQA Clue | 75.9  | 51.9      | 65.3     | 47.3      |
> | Caption + LLM Clue | 72.9  | 58.1      | 57.1     | 52.9      |
>
>   As shown above, the conclusions drawn from Table 1, and the strategy remain unchanged compared with the results on 500 examples.
>
>
> **[The rationale for branching based on the problem category and the confidence]**
>
> In the decision of choosing the branch for solving the question, we consider two aspects: performance and efficiency, evaluating by the number of LLM calls.
>
> **(a)** From Table 1, we can observe that when the LLM is confident about its initial reasoning, its performance is superior or nearly the best for both VCU and VCI problems. In addition, [LLM w/ Caption] is the most compute-efficient. Therefore, using [LLM w/ Caption] is a **cost-effective** option.
>
> **(b)** When the LLM is not confident about its initial reasoning on VCI problems, [LLM w/ Caption+LLM clue] significantly outperforms other decision paradigms. On VCU problems, we can observe that the performance of BLIP2 is similar to [LLM w/ Caption+LLM clue]. However, the [LLM w/ Caption+LLM clue] requires five LLM calls, which is three LLM calls more than using BLIP2. Therefore, using BLIP2-ITA is a **cost-effective** option in this case.
>
> We briefly mentioned the efficiency aspect in the original version. We’ve updated the explanation for the branching rationale in Section 6.1.
>
> **[Why use words to transfer information between LLMs and VLMs]**
>
> We acknowledge that relying solely on text for communication between LLMs and VLMs is a limitation of our study, a point we also mentioned in the limitations section. However, we believe our submission is a valuable addition to the literature due to following reasons:
>
> **(a)** Currently, connecting top-performing closed-source proprietary LLMs directly with perception (e.g. image, video, point cloud, etc.) modules through embeddings is challenging. As a result, the use of captions remains the only method for establishing this connection. Hence, most current research in multi-modal applications of LLMs primarily relies on caption-based approaches.
>
> **(b)** Utilizing text for communication between LLMs and VLMs provides a clear advantage in terms of interpretability. This interpretability is crucial for comprehending the behavior of our method. While embeddings could serve as an alternative medium, they tend to act as a "black-box", offering limited interpretability. This lack of clarity can lead to challenges in debugging and elongate the development process.
>
> **(c)** In our initial investigations, we observed a comparable effect when linking LLMs and VLMs through embeddings. Specifically, these models can neglect important visual cues in scenes characterized by clutter and occlusions because they do not actively search for visual cues based on the context of the scene and the question. One possible extension of our approach could involve implementing an attention mechanism over visual embeddings, guided by a high-level comprehension of the scene and contextual information, much like the methodology we have employed.

---

### Comment · Area_Chair_WyaX · 2023-11-22

Hi Reviewers for paper 4716,

The authors have responded to your reviews. Pls read and reply to them.

Thanks,
AC

---

### Meta-Review · Area_Chair_WyaX · 2023-12-05

**Metareview:**

This was an interesting paper, with reviewers finding the strengths to be its good motivation, usefulness and novelty of the framework for LLMs and VLMs to "collaborate". Personally, this AC finds the premise of the work to be very interesting, perhaps with a significance even beyond what the authors present or realize.

I find the biggest takeaway of lasting significance to be that: whereas ML/DL papers are classically about individual models on individual problems (e.g. AlexNet on classification [ImageNet]), the community should seriously consider whether this paradigm has run its course. "Modern" problems such as Visual Commonsense Reasoning, while sometimes simplistically treated as one problem, are clearly a collection of diverse, inter-connected sub-problems, each of which are likely best tackled by different algorithms or approaches. This paper demonstrates that by recognizing that sub-problems (e.g. "VCU" and "VCI") exist, and designing a framework for different approaches (VLMs and LLMs) to work together differently depending on the sub-problems, performance can improve.

However, much as I like the premise, after much consideration of the reviews, rebuttals and further responses, I do not recommend acceptance of this work in its current form, for the reasons explained below. Both qualitatively and quantitatively, the reviewers' lukewarm, slightly positive-leaning regard -- even post-rebuttal -- are consistent with this. (Important note: one reviewer appears to have seriously misunderstood the paper, and I have fully taken this into account.)

The generality and robustness of the evaluations need to be significantly improved, even after extra results in the rebuttal (e.g. increase from 500 to 3000 samples from the VCR dataset). For instance, 3000 is still barely above 10% of the VCR validation set, and 1000 samples from OK-VQA is less than 10% of 14000 total. The authors did not provide any strong or convincing justification for why such a small sampling was done, or even why 100% of the questions couldn't have been used. Furthermore, achieving SOTA results aside, the work would have been more convincing and impactful had a larger set of LLMs and VLMs been used to make a larger point about a framework for LLMs/VLMs to work together.

Furthermore, the overall framing could be significantly improved, which would also avoid certain (reasonable) misunderstandings or disagreements. For instance, the abstract and introduction mention the division of VCR into VCU and VCI as part of the problem premise. Readers can reasonably disagree about this division. Instead, if the division into VCU and VCI were framed as just one possible division as part of the proposed method (rather than the problem definition), then the positive results can justify this division.

Overall, for a highly selective/competitive venue like ICLR, the work in its current form does not justify acceptance.

**Justification For Why Not Higher Score:**

The evaluation aspects of the paper need to be significantly improved.

**Justification For Why Not Lower Score:**

N/A

---

### Decision · Program_Chairs · 2024-01-16

Reject